# JAK-STAT Pathway Regulation of Intestinal Permeability: Pathogenic Roles and Therapeutic Opportunities in Inflammatory Bowel Disease

**DOI:** 10.3390/ph14090840

**Published:** 2021-08-25

**Authors:** Hillmin Lei, Meli’sa S. Crawford, Declan F. McCole

**Affiliations:** Division of Biomedical Sciences, School of Medicine, University of California, Riverside, CA 92521, USA; hlei013@ucr.edu (H.L.); melisac@ucr.edu (M.S.C.)

**Keywords:** tight junctions, intestinal permeability, inflammatory bowel disease, JAK-STAT

## Abstract

The epithelial barrier forms the interface between luminal microbes and the host immune system and is the first site of exposure to many of the environmental factors that trigger disease activity in chronic inflammatory bowel disease (IBD). Disruption of the epithelial barrier, in the form of increased intestinal permeability, is a feature of IBD and other inflammatory diseases, including celiac disease and type 1 diabetes. Variants in genes that regulate or belong to the JAK-STAT signaling pathway are associated with IBD risk. Inhibitors of the JAK-STAT pathway are now effective therapeutic options in IBD. This review will discuss emerging evidence that JAK inhibitors can be used to improve defects in intestinal permeability and how this plays a key role in resolving intestinal inflammation.

## 1. Introduction

Inflammatory bowel diseases (IBD) are chronic inflammatory disorders of the gastrointestinal tract and primarily include Crohn’s disease (CD) and ulcerative colitis (UC). Inflammation in CD predominantly occurs in the small intestine while UC is commonly associated with inflammation of the rectum and areas of the colon [1,2,3]. Being a transmural disease, the presence of granulomas, focal crypt architectural abnormalities, and mucin preservation at active sites are frequently seen in CD patients [4]. However, focal or diffuse basal plasmacytosis, widespread mucosal or crypt architectural distortion, and mucin depletion are commonly seen in UC patients [4,5,6]. Collectively, patients with IBD can present a multitude of symptoms including diarrhea, abdominal pain, weight loss and in severe cases, nutrient deficiency, and anemia [4]. Furthermore, patients with IBD are at risk of developing complications including intestinal obstruction and colorectal cancer that can result in hospitalization and subsequent surgical interventions [4]. Despite the abundance of therapeutic strategies for IBD, many patients fail to respond appropriately to the available treatments [1,3]. Moreover, the prevalence of IBD is consistently rising with approximately 7 million cases globally [7].

Inflammatory bowel diseases are multifactorial disorders that manifest from an abnormal immune response towards luminal contents in the digestive tract in processes driven by genetic and environmental risk factors [8,9]. Currently, over two hundred SNPs have been linked to the pathogenesis of IBD, many of which are involved in regulating intestinal epithelial barrier function [8]. Disruption of the intestinal epithelial barrier, via increased intestinal permeability, is a feature of many chronic inflammatory disorders including IBD. However, it was uncertain whether the increased intestinal permeability found in patients with CD or UC is an associated finding or if it plays a significant role in disease pathogenesis [10].

Intriguingly, a prospective study by Turpin et al. demonstrated that increased intestinal permeability is associated with the later development of CD [11]. This finding has substantial implications in understanding the pathogenesis of IBD as it provides the most robust evidence of increased intestinal permeability predisposing to subsequent onset of CD. Variants in genes belonging to the Janus kinase (JAK)—signal transducer and activator of transcription (STAT) signaling pathway, as well as members of the protein tyrosine phosphatase (PTP) family that act as negative regulators of JAK-STATs, are associated with increased IBD risk [8,9]. JAK-STAT proteins are responsible for mediating receptor signaling of numerous IBD-associated cytokines involved in regulating intestinal permeability [9]. With regulation of intestinal permeability being an essential component of proper intestinal epithelial barrier function, developing therapeutics to target signaling cascades such as the JAK-STAT pathway represent an attractive therapeutic avenue for patients with IBD. 

Indeed, inhibitors of the JAK-STAT pathway have been an emerging therapeutic option in IBD (Table 1) [12]. Unlike biologic treatments for IBD such as tumor necrosis factor (TNF) inhibitors, JAK inhibitors are small molecule drugs that have a brief plasma half-life, low risk of immunogenicity, and rapid onset of action [13]. Tofacitinib (Xeljanz^TM^), the first of its class to be approved for the treatment of moderate and severe UC, demonstrated efficacy of remission in UC patients; however, it showed no significant efficacy in induction and maintenance of Crohn’s disease activity index (CDAI; <150) vs. placebo in phase 2b clinical trials [14]. The differential effects of the broad spectrum JAK inhibitor tofacitinib vs. the selective JAK1 inhibitors in treating UC vs. CD respectively are noteworthy and may indicate a more selective role for JAK1 in the pathogenesis of CD. Treatment regimens for UC that utilize tofacitinib are primarily for patients that failed conventional therapies and/or lost responsiveness to other biologics [15]. However, drugs that inhibit the JAK-STAT pathway may impact other physiological processes thus requiring additional long-term studies regarding their safety profiles and mechanism of action [13]. Of note, tofacitinib was allocated a ‘boxed warning’ label by the FDA due to blood clotting events associated with the 10-mg, twice-daily dose of tofacitinib when administered to rheumatoid arthritis patients with at least one cardiovascular risk factor and was associated with a risk of infections [16]. Therefore, understanding the role of the JAK-STAT pathway in regulating physiological processes such as maintenance of intestinal permeability is crucial for developing effective therapeutic agents to reinforce the intestinal epithelial barrier and intestinal homeostasis. 

Here, we will review emerging evidence of the importance of JAK-STAT signaling in the regulation of intestinal permeability and associated pathologies, and how JAK inhibitors can be used to improve defects in intestinal permeability and resolve intestinal inflammation. 

## 2. Intestinal Permeability in IBD

### 2.1. Intestinal Epithelial Barrier

The intestinal epithelial barrier is a single layer of epithelial cells that is essential for regulation of intestinal homeostasis and mediates communication between the intestinal microflora and the immune system [17,18]. In addition to being a dynamic interface between the luminal contents and the host immune system, the epithelium is the first site of exposure to many of the environmental factors that can trigger disease activity in chronic IBD [17,18]. With the epithelium neighboring the luminal microbes and the underlying immune system, the integrity of the epithelial lining is critical to avoid excessive contact of pathogenic antigens with lamina propria immune cells. Aside from serving as a physical barrier, intestinal epithelial cells can also endocytose bacteria, sequester and neutralize toxins, and detect pathogen-associated molecular patterns (PAMPs) [19]. Furthermore, the epithelium can respond to intestinal damage by secreting factors that contribute to epithelial restitution, initiating wound repair, and activating the underlying innate and adaptive immunity [18]. However, disruption of this selective barrier can lead to chronic intestinal inflammation through the uncontrolled uptake of foreign antigens and the overstimulation of the mucosal immune system [20]. Along with its innate immune functions, the intestinal epithelium can perform a variety of specialized tasks due to its ability to form a tightly regulated and selectively permeable barrier. This leaky barrier is essential for the epithelium to generate ion solute concentration gradients to appropriately absorb nutrients and water [18]. Moreover, a regulated intestinal barrier is required for controlled antigen transport to the resident immune cells in the gut-associated lymphoid tissue, thus supporting the maturation of the immune system [21]. Therefore, proper function of the intestinal epithelial barrier is essential for prevention of intraluminal pathogen invasion, regulated intake of essential nutrients, and managing the intestinal immune response.

### 2.2. Intestinal Permeability: Maintenance and Its Role in Inflammation

An impaired intestinal epithelial barrier, reflecting increased intestinal permeability, has been linked to several diseases including celiac disease, type 1 diabetes, and IBD. However, a clear connection between elevated underlying intestinal permeability and the onset of IBD had not been made until a recent prospective study of a cohort of asymptomatic first-degree relatives of patients with IBD reported that elevated intestinal permeability is significantly linked to the risk of developing CD [11]. Baseline measures of the urinary fractional excretion ratio of lactulose to mannitol (LMR) in subjects who later developed CD were significantly higher than asymptomatic individuals [11]. The manifestation of disease did not occur until several years after the initial identification of increased permeability thus suggesting that increased permeability was required, but by itself was not sufficient to cause disease [11]. In addition, a genome-wide association of LMR determined that host genetics provide a mild contribution to an elevated LMR [22]. Further investigations are needed to determine if the abnormality of the gut barrier is an intrinsic barrier defect, the result of a specific insult, or a reflection of the luminal environment [11]. These findings represent a major advance on earlier work that alluded to a primary role for increased permeability in the onset of IBD [23,24,25].

These clinical findings align with studies in animal models of IBD demonstrating that increased intestinal paracellular permeability precedes the onset of inflammation in the gut. The SAMP1/YitFc mouse model of chronic ileitis was described to have a permeability defect before the presence of ileal inflammation [26]. SAMP mice exhibited profound epithelial barrier alterations with reduced transepithelial electrical resistance (TEER) and elevated LMR [26]. Similarly, *mdr1a*^−/−^ mice, lacking the multidrug resistance gene that encodes the p-glycoprotein transporter, presented with chronic inflammation of the gut when exposed to a “normal” microbiota [27]. A defect in *mdr1a* is likely to predispose the intestinal epithelium to bacterial invasion and infection [27]. Madsen et al., showed that increased intestinal permeability is evident before intestinal inflammation in *Il10*^−/−^ mice [28]. Of interest, treatment of *Il10*^−/−^ mice with the probiotic compound, VSL#3, results in the normalization of colonic epithelial barrier along with a reduction in levels of pro-inflammatory cytokines including TNF-α and IFN-γ [28]. 

Mechanistically, alterations in expression of tight junction proteins have been reported to affect intestinal permeability and contribute to the pathogenesis of IBD. Mice with intestinal deletion of claudin-7 exhibited colonic inflammation even though tight junction structures were intact, suggesting differences in expression of certain tight junction proteins can increase or reduce intestinal paracellular permeability [29]. Similar effects were also seen in colonic tissues from patients with active CD. These patients showed impaired intestinal barrier function as indicated by a dramatic reduction in TEER [30]. Moreover, this drop in intestinal barrier function was associated with a decrease in tight junction strand number and continuity, in addition to an increase in epithelial apoptosis [30]. The size of the discontinuity of these tight junction strand breaks were considered large enough to facilitate paracellular passage of foreign antigens [30]. Expression of the barrier sealing tight junction proteins, occluding, and claudins -3, -5, and -8, were also dramatically decreased in patients with active CD while increased expression of the pore-forming claudin-2 was upregulated [30].

Consistent with the clinical findings of Turpin et al., animal studies also indicate that increased intestinal paracellular permeability by itself is insufficient to cause colitis [11]. Su et al., demonstrated that constitutively-active myosin light chain kinase transgenic mice had significant barrier loss but grew similarly to their gender-matched littermates and did not develop intestinal inflammation [31]. However, when colitis was induced in these mice, they suffered from more severe inflammation and had significantly shorter survival times [31]. Furthermore, junctional adhesion molecule-A (JAM-A) deficient mice had normal intestinal epithelial architecture in their colonic mucosa despite the increased presence of lymphoid aggregates and reduced intestinal barrier function [32]. Intriguingly, these JAM-A deficient mice were more sensitive to dextran sulfate sodium (DSS)-induced colitis [32,33]. Overall, these studies support the emerging hypothesis that an impaired barrier defect along with a trigger of an inappropriate immune response is critical for intestinal inflammation. Nevertheless, maintaining intestinal barrier function via regulating tight junction proteins is necessary for preventing intestinal inflammation (Figure 1) and developing therapeutics that target their regulators may be a viable clinical approach in patients with IBD.

## 3. Protection of the Intestinal Epithelial Barrier

The intestinal epithelial barrier consists of multiple elements that contribute to its function as a physical, chemical, and immunological defense. While the mucus layer, intestinal epithelium, and the underlying immune cells in the lamina propria comprise a major component of the defense response, intestinal epithelial permeability is regulated by tight junctions. In the following section, we will discuss the role and mechanisms of tight junctions as a selectively permeable barrier, and the significance of the JAK-STAT pathway in barrier permeability.

## 4. Pore Pathway: Electrolyte Flux

### Claudins

Permeability through the tight junction barrier consists of two functionally distinct events that can be distinguished both functionally and on a molecular level from “unrestricted” permeability arising from cell death [34,35]. First, there is the high-capacity, charge-selective, and size-selective pore pathway that is predominantly regulated by the expression of claudin proteins, which permits the passage of small ions and uncharged molecules (Figure 2) [17,20,36,37]. Claudins are a large family (>24 members) of transmembrane proteins that are essential for managing paracellular transport of ions and solutes.

Dysregulation of claudins has been associated with an increase in epithelial barrier permeability and the development of IBD. As previously discussed, patients with active CD show reduced expression of the barrier-sealing claudins, -3, -5 and -8 while having increased expression of the pore-forming claudin-2 [30]. Similarly, in another cohort of patients with IBD, expression of claudin-2 was also upregulated whereas the expression of claudin-3 and claudin-4 were noticeably decreased [38]. Li et al., demonstrated a critical role of the dynamin-dependent endocytosis of claudin-3 and claudin-4 under nutrient stress in IECs [39]. These changes combine to accentuate tight junction permeability to cations and water thereby contributing to diarrhea, one of the major clinical symptoms of IBD [40].

Interestingly, IBD-associated pro-inflammatory cytokines including IL-6, IL-22 and IFN-γ have been reported to activate the JAK-STAT pathway to modify expression of claudin proteins [9,35,41,42,43,44]. Elevated claudin-2 expression in IL-6 treated Caco-2 monolayers is negated with the addition of STAT3 siRNA [35]. Two potent inhibitors of JAKs, JAK inhibitor I and AZD1480, reversed the upregulation of *CLDN2* transcription and claudin-2 protein by abrogating the phosphorylation of STAT1 and STAT3 in IL-22 treated Caco-2 monolayers [41]. Tofacitinib, a pan-JAK inhibitor that is FDA-approved to treat moderate to severe UC, is able to restrict IFN-γ-induced claudin-2 promoter activity and the increase in claudin-2 protein levels in T_84_ IECs by significantly reducing activation of JAK1-STAT1/3 signaling [9]. The induction of claudin-2 by IFN-γ has also been shown to be mediated by STAT1 activation and binding to the *CLDN2* promoter for transcriptional expression [44]. This regulatory pathway was accentuated by knockdown of PTPN2 in vitro and in mouse intestinal epithelium in vivo [44,45]. *PTPN2* is a key negative regulator of the JAK-STAT pathway and a loss-of-function variants in this gene increase the risk of IBD onset [46,47,48,49]. Importantly, the decreased TEER arising from PTPN2 knockdown in vitro was reversed by transfection with claudin-2 siRNA thus identifying a critical role for this negative regulator of JAK-STAT signaling in regulating claudin-2 transcriptional regulation [44,45]. A second mechanism of PTPN2 regulation of claudin-2 was also identified in which loss of PTPN2 reduced expression of the serine protease, matriptase (*ST14*) [50]. Matriptase mediates removal of claudin-2 from tight junctions and *St14* hypomorphic mice exhibit increased claudin-2 expression and reduced TEER [50].

## 5. Leak Pathway: Molecular Mediators

### 5.1. Zonula Occludens

The tight junction-mediated low-capacity leak pathway allows the paracellular passage of larger ions and molecules (<100 Å) regardless of charge (Figure 3) [20]. The conglomeration of proteins that generally mediate the leak pathway include zonula occludens (ZO), occludin, tricellulin, and JAM-A. ZO is a PSD95-DLG1-ZO-1 homology domain (PDZ)-containing intracellular plaque protein involved in forming a scaffold between transmembrane proteins and the actin cytoskeleton [51,52]. At the tight junction, ZO-1, -2, and -3 contain PDZ domains that facilitate clustering and anchoring of tight junction proteins including occludin, claudins, and JAM-A to the cytoplasm [53,54,55].

In CD patients, ZO-1, which is normally found in the apical portion of IECs at tight junctions, was relocated to the basolateral side and were also found in the lamina propria extracellular matrix [56]. Intriguingly, the cytoskeletal architecture, represented by F-actin, was maintained, further supporting the notion that inappropriate localization of tight junction proteins may contribute to increased intestinal permeability seen in patients with IBD [56]. Gassler et al., demonstrated that expression of ZO-1, but not ZO-2, was downregulated in inflamed mucosal tissue of active IBD patients, suggesting both specific and broad functions of ZOs in the establishment of the tight junction network in a normal and inflamed mucosa [57]. Reduced expression of ZO-1 was also frequently seen in patients with active CD and UC, while alterations in ZO-1 function and location are likely mediated by cytokines released during intestinal inflammation [58].

Similar to the claudin proteins, the JAK-STAT signaling pathway is also involved in the regulation of ZO-1. Knockdown of *PTPN2* in human Caco-2BBe IECs induced the internalization of ZO-1 and formation of gaps between adjacent IECs [8]. Loss of *Ptpn2* in macrophages of mice caused a reduced and more diffuse staining of ZO-1 in the colonic epithelium, demonstrating a major role of these immune cells in regulating intestinal barrier function [8]. Of interest, these effects were nullified with the addition of tofacitinib [8]. Correspondingly, tofacitinib was able to prevent the relocalization of ZO-1 and minimize the number of intercellular junctional gaps in IFN-γ-treated T_84_ IECs [9]. In human colonic organoids, tofacitinib also reversed the IFN-γ-induced increase in 4-kilodalton fluorescein isothiocyanate-dextran (FD4) influx, a measurement of macromolecular paracellular permeability [9]. 

### 5.2. Occludin

Occludin is an integral transmembrane protein that is involved in organizing and stabilizing tight junctions and was the first identified component of the tight junction strand [59,60]. Downregulation and redistribution of the levels of occludin from tight junctions have been frequently reported in UC and CD [30,61,62]. Occludin has also been shown to be regulated in a JAK-STAT dependent manner. Moreover, occludin expression in IFN-γ-treated T_84_ cell monolayers was not significantly altered although intercellular gaps in membrane localization of ZO-1—in the apical membrane region as indicated by confocal z-stack imaging of apical occluding—was observed [9]. While tofacitinib was able to reduce the number of these gaps, further investigations are needed to determine the role of occludin in these events [9]. Interestingly, decreased expression of occludin in cultured *PTPN2*-deficient IECs and in proximal colon IECs from *Ptpn2*-LysMCre mice was rescued with tofacitinib [8]. The correction of occludin levels by tofacitinib was associated with normalization of paracellular permeability [8]. Moreover, a recent study from our lab demonstrated that loss of *Ptpn2* in mice in vivo disrupted tight junction localization of occludin and ZO-1 in colonic epithelium [45]. Further studies are required to determine which cytokines regulate occludin expression and localization in the gut mucosa.

### 5.3. Tricellulin

Tricellulin is a tight junction protein that forms a barrier against macromolecules and solutes in bicellular and tricellular tight junctions [63]. Krug et al., demonstrated that tricellulin expression was reduced in the sigmoid colon of patients with UC and likely contributes to the enhanced permeability of macromolecules [64]. These findings were confirmed in a study showing that lower expression of tricellulin and the associated defects were reversed during ulcerative colitis remission [65]. 

Notably, downregulation of tricellulin in UC was shown to be driven by the UC-associated cytokine IL-13 through IL-13 receptor α2 [66]. IL-13-induced suppression of tricellulin in the intestinal epithelial cell line HT-29/B6 resulted in increased FD4 permeability [66]. Interestingly, inhibitors targeting JAK1 or JAK2 prevented the reduction in tricellulin levels by IL-13 [66]. In contrast, expression of tricellulin was unaltered in IFN-γ-treated-human colonic organoids or T84 monolayers [9]. Reduction of tricellulin levels was also seen when control Caco-2BBe cells were co-cultured with *PTPN2*-deficient macrophages, and this was rescued by incubation with anti-IL6 antibody, thus confirming a role for immune cell secreted IL-6 in causing decreased tricellulin expression in neighboring IECs [8,35]. Tofacitinib treatment of control Caco-2BBe cells, but not *PTPN2*-deficient IECs, rescued the reduction in tricellulin expression [8].

In a recent study, localization of tricellulin was found to be shifted predominantly from intestinal crypts to the surface epithelium in CD patients [66]. A key regulator of tricellulin localization, angulin-1, was also discovered to be downregulated in active CD [67]. Intriguingly, leptin, a hormone that is primarily produced by adipose tissue to regulate appetite and food storage, is able to downregulate angulin-1 levels in T84 and Caco-2 cell lines [67]. Adipose tissue is commonly seen adjacent to the inflamed intestinal segments in CD and could be a potential source of leptin [68]. Caco-2 cells treated with leptin exhibited increased intestinal permeability as demonstrated by elevated FD4 flux [67]. These effects were negated by pre-treatment with inhibitors of STAT3, Stattic and WP1006, and partially by the JAK2 inhibitor AG490 [67]. These findings reveal leptin as a novel target for the development of JAK-STAT inhibitors to alleviate defects in the intestinal barrier.

### 5.4. JAM-A

JAM-A is another PDZ domain-containing integral membrane protein that is selectively concentrated at intercellular junctions of epithelial cells [69]. Kucharzik et al., reported that JAM-A expression was decreased in the inflamed mucosa of patients with active IBD [70]. This finding was confirmed in a later study showing that normal colonic mucosa expressed abundant epithelial JAM-A; however, in the mucosa of both UC and CD its expression was reduced extensively in inflamed tissue [33]. 

Intriguingly, Fan et al., reported that the cytoplasmic tail of JAM-A is tyrosine phosphorylated (p-Y280) in association with impaired intestinal barrier function [71]. Exposure of human IECs to pro-inflammatory cytokines including TNF-α, IFN-γ, IL-22, or IL-17A resulted in increased JAM-A p-Y280 [71]. Furthermore, IL-17A enhanced transepithelial flux of FD4; however, there was only a mild reduction in TEER [71]. The regulation of this tyrosine phosphorylation was associated with the Src kinase, YES-1, and the phosphatase, PTPN13, suggesting the barrier sealing function of JAM-A is dependent on tyrosine kinase and phosphatase activity [71].

Along with the above findings, the JAK-STAT pathway has been identified to be a regulator of JAM-A. *PTPN2*-deficient IECs co-cultured with *PTPN2*-deficient macrophages have significantly reduced expression of JAM-A; however, tofacitinib was unable to restore the levels of JAM-A [8,35]. In contrast, JAM-A levels were corrected by tofacitinib treatment in proximal colon IECs from *Ptpn2*LysMCre mice [8]. Collectively, these studies indicate that JAM-A regulation is altered under inflammatory conditions and contributes to the increased permeability seen in IBD patients. Table 2 summarizes the regulation of tight junction proteins via the JAK-STAT pathway.

## 6. JAK-STAT Signaling in Apoptosis and Necroptosis of Intestinal Epithelial Cells

Along with its vital role in regulating intestinal permeability via tight junction proteins, epithelial barrier function can be altered via the “unrestricted pathway”, which is primarily associated with cell loss due to death or shedding [72]. This can also be modulated by JAK-STATs as JAK-STAT signaling can mediate different forms of cell death, apoptosis, and necroptosis in IECs. Deletion of *Stat1* in mouse ileal IECs protected against IFN-λ-induced necroptosis and apoptosis in Paneth cells [73]. When treated with tofacitinib, IFN-λ-treated small intestinal organoids had lower gene transcription of the cell death mediators, *Mlkl* and *Caspase*-*8* [73]. Notably, the addition of tofacitinib restored the viability of the organoids and prevented Paneth cell death in response to IFN-λ [73]. Filgotinib, a novel selective JAK1 inhibitor, showed similar effects and interfered with STAT1 phosphorylation [73].

In a related study, deletion of *Stat1* in mouse ileal IECs lacking *Caspase*-*8* partially recovered Paneth cell numbers and was associated with reduced *Mlkl* expression and a lower number of TUNEL (terminal deoxynucleotide transferase dUTP nick end labeling) positive cells [74]. In contrast, Paneth cell function did not return to normal levels, indicating an alternative mechanism that regulates Paneth cell activity [74]. Surprisingly, *Stat1* was demonstrated to be unessential in regulating cell death in the colon when inflammation was induced with DSS [74].

Of interest, an increase of reserve stem cells was observed after treatment of enteroids and mice with TNF-α and IFN-γ, suggesting cytokines released during mucosal inflammation are required for the regenerative response [75]. Notably, the JAK-STAT1 pathway was found to be necessary for activation of reserve stem cells during inflammation as pre-treatment of enteroid cultures with tofacitinib prevented the IFN-γ-induced increase in reserve stem cell number [75]. These findings demonstrate an alternative role of the JAK-STAT pathway in facilitating the restitution of the epithelial barrier.

## 7. Conclusions

In summary, the intestinal epithelial barrier is the first site of exposure to many environmental agents that can trigger disease manifestation in chronic IBD. Alterations in the epithelial barrier that increase intestinal permeability are a key feature of chronic inflammatory disorders including celiac disease, type 1 diabetes, and IBD. Recent studies have demonstrated the importance of the JAK-STAT signaling pathway in the pathogenesis of IBD by mediating cytokine-induced changes in intestinal paracellular permeability through regulating tight junction protein expression and localization. Additionally, the JAK-STAT signaling pathway can alter epithelial barrier function by regulating cell death in the intestinal epithelium. Moreover, the JAK-STAT signaling pathway is involved in processes related to immune system regulation and cellular development.

Currently, inhibitors of the JAK-STAT pathway represent a promising therapeutic option for the treatment for IBD; however, these compounds have potential risks including non-specificity, toxicity, and efficacy. While much of the focus of JAK inhibitors such as tofacitinib has centered on their effects on immune cells, emerging evidence shows that they can directly affect epithelial cell pathways regulating tight junction proteins. Furthermore, there are many therapeutic strategies that target inflammatory pathways associated with IBD, yet there are no current treatments for restoring the intestinal barrier. Thus, further understanding of the direct influence(s) of the JAK-STAT pathway on tight junction proteins and their upstream regulators are warranted. Specific inhibition of downstream targets of the JAK-STAT pathway may present an attractive strategy to develop novel and effective therapeutics focused on barrier regulation. Such investigations will be critical to better understand the scope of current and emerging JAK inhibitors to modulate cell types and pathways capable of modulating intestinal permeability and generate further insight into the clinical benefit of this class of agents. 

## Figures and Tables

**Figure 1 pharmaceuticals-14-00840-f001:**
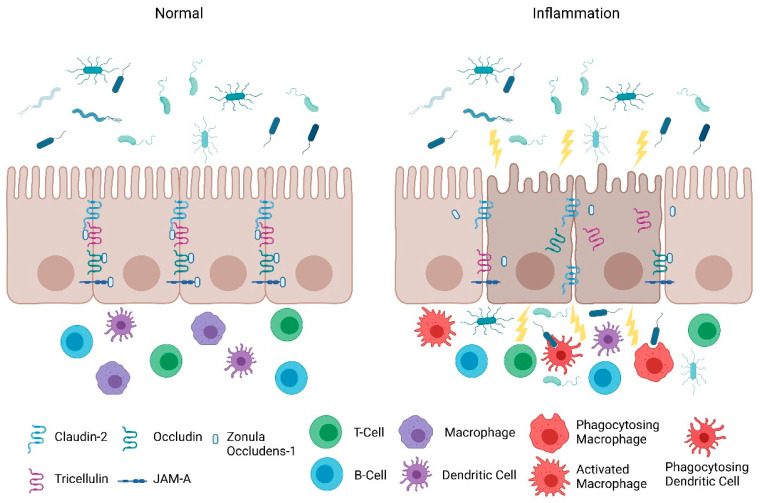
**Intestinal permeability leads to intestinal inflammation.** Tight junctions are the primary regulator of intestinal paracellular permeability and act as a selective ‘gate’ for nutrients and fluids while preventing passage of foreign antigens. Alterations in expression and localization of tight junction proteins can lead to increased intestinal permeability and provide a pathway for paracellular access of luminal pathogenic agents to lamina propria immune cells, in addition to loss of electrolytes and fluid. Accumulation of these agents can potentially overwhelm the underlying immune system and collectively cause intestinal inflammation. This illustration was made using BioRender.

**Figure 2 pharmaceuticals-14-00840-f002:**
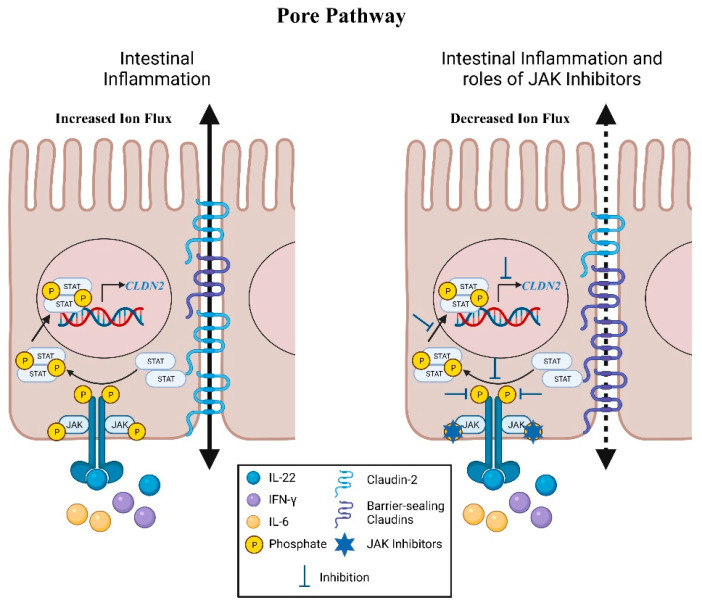
**JAK-STAT regulation of intestinal permeability: Pore Pathway.** During intestinal inflammation, proinflammatory cytokines are released and bind to their respective receptors on intestinal epithelial cells. The associated cellular tyrosine kinases, JAKs, are then brought together and activated. These interactions subsequently lead to phosphorylation of the cytokine receptor which act as hubs for STATs. The JAKs would then phosphorylate the STATS and the phosphorylated STATs would dimerize, enter the nucleus, and serve as transcription factors for genes that express tight junction proteins which influence intestinal permeability. In the pore pathway, claudin-2 expression is upregulated in response to IL-22, IFN-γ, and IL-6. Expression of barrier-sealing claudins is reduced; however, the mechanisms that underlie these alterations require further investigation. The JAK inhibitors tofacitinib, AZD1480, and JAK inhibitor I, reduce expression of claudin-2 and consequent paracellular permeability via the pore pathway. Downregulation of claudin-2 expression has also been reported with STAT1 and STAT3 siRNA intervention. Levels of barrier-sealing claudins are suggested to return to normal levels during remission of inflammation. These illustrations were made using BioRender. (*CLDN2*: Claudin-2 Gene, JAK: Janus Kinase, STAT: Signal Transducer and Activator of Transcription).

**Figure 3 pharmaceuticals-14-00840-f003:**
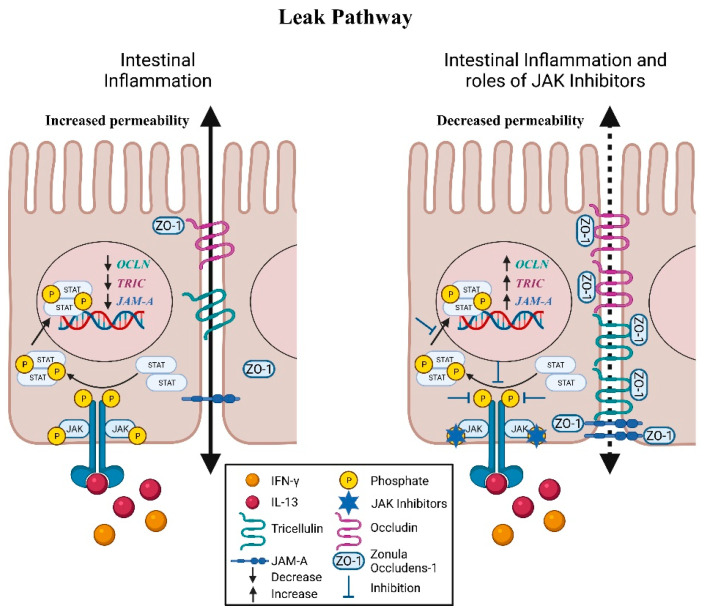
**JAK-STAT regulation of intestinal permeability: Leak Pathway.** The leak pathway is associated with altered expression and localization of ZO-1, occludin, and tricellulin resulting in lower levels of these proteins at the tight junction. Expression of JAM-A is also downregulated through the JAK-STAT pathway. JAK1 and JAK2 inhibitors prevented the decrease in tricellulin expression by IL-13. Furthermore, with tofacitinib treatment, the reduced expression and mislocalization of ZO-1 and occludin induced by IFN-γ are mostly reversed. JAM-A levels were also corrected by tofacitinib. These illustrations were made using BioRender. (JAK: Janus Kinase, STAT: Signal Transducer and Activator of Transcription, OCLN: Occludin Gene, TRIC: Tricellulin Gene, JAM-A: Junctional Adhesion Molecule-A Gene).

**Table 1 pharmaceuticals-14-00840-t001:** Current clinical trials of JAK Inhibitors in IBD patients. **Source:** clinicaltrials.gov. **Accessed:** 5 August 2021.

JAK Inhibitors	Target	Study	Administration	Phase
SHR0302	JAK1	A Phase II Study in Patients With Moderate to Severe Active Crohn’s Disease	N/A	Phase 2
TD-1473	Pan-JAK	Efficacy and Safety of TD-1473 in Crohn’s Disease	Oral	Phase 2
TD-1473	Pan-JAK	TD-1473 Long-Term Safety (LTS) Ulcerative Colitis (UC) Study	Oral	Phase 2/Phase 3
TD-1473	Pan-JAK	Efficacy and Safety of TD-1473 in Ulcerative Colitis	Oral	Phase 2B/Phase 3
Upadacitinib	JAK1	A Maintenance and Long-Term Extension Study of the Efficacy and Safety of Upadacitinib (ABT-494) in Participants with Crohn’s Disease Who Completed the Studies M14-431 or M14-433	Oral	Phase 3
Upadacitinib	JAK1	A Study to Evaluate the Long-Term Safety and Efficacy of Upadacitinib (ABT-494) in Participants with Ulcerative Colitis (UC)	Oral	Phase 3
Upadacitinib	JAK1	A Study of the Efficacy and Safety of Upadacitinib (ABT-494) in Participants with Moderately to Severely Active Crohn’s Disease Who Have Inadequately Responded to or Are Intolerant to Biologic Therapy	Oral	Phase 3
Upadacitinib	JAK1	A Study to Evaluate the Safety and Efficacy of Upadacitinib (ABT-494) for Induction and Maintenance Therapy in Participants With Moderately to Severely Active Ulcerative Colitis (UC)	Oral	Phase 3
Upadacitinib	JAK1	A Study of the Efficacy and Safety of Upadacitinib (ABT-494) in Participants With Moderately to Severely Active Crohn’s Disease Who Have Inadequately Responded to or Are Intolerant to Conventional and/or Biologic Therapies	Oral	Phase 3
Upadacitinib	JAK1	A Study to Evaluate the Long-Term Efficacy, Safety, and Tolerability of Repeated Administration of Upadacitinib (ABT-494) in Participants with Crohn’s Disease	Oral	Phase 2
Tofacitinib	Pan-JAK	A Study of Tofacitinib in Patients with Ulcerative Colitis in Stable Remission	Oral	Phase 4

**Table 2 pharmaceuticals-14-00840-t002:** JAK-STAT signaling pathways affecting intestinal permeability. (JAK: Janus Kinase, STAT: Signal Transducer and Activator of Transcription, IL-22: Interleukin 22, IFN-γ: Interferon gamma, IL-6: Interleukin 6, IL-13: Interleukin 13, JAM-A: Junctional Adhesion Molecule-A, ZO-1: Zonula Occludens-1, TEER: Transepithelial electrical resistance, FD4: 4-kilodalton fluorescein isothiocyanate-dextran, SOCS: Suppressor of Cytokine Signaling, siRNA: Small interfering ribonucleic acid).

JAK-STATs	Activators in IBD	Effect on TJ Proteins	Effect on Permeability	JAK/STAT Inhibitors	References
STAT1 and STAT3	IL-22	Increased claudin-2 expression	Increased paracellular permeability to ionic solutes; Reduced TEER	JAK Inhibitor 1 and AZD1480 (Inhibited the STAT3-dependent gene, *SOCS3*)	[41]
JAK1-STAT1/STAT3	IFN-γ	Increased claudin-2 expression	Increased paracellular permeability; Reduced TEER, Increased FD4 permeability	Tofacitinib	[9]
STAT3	IL-6	Increases claudin-2 expression	Increases paracellular permeability to ionic solutes; Reduced TEER	AG490, STAT3 siRNA	[35,43]
Undetermined	Presumably IFN-γ	Decreased JAM-A expression; Possible redistribution of JAM-A	Increased paracellular permeability to macromolecules; Presumably reduces TEER and increases FD4 permeability	Tofacitinib	[8]
Undetermined	IFN-γ	Decreased occludin expression; Redistribution of occludin	Increased paracellular permeability to larger macromolecules	Tofacitinib	[8,9]
JAK1/JAK2	IL-13, IFN-γ	Downregulation of tricellulin; Redistribution of tricellulin	Increased uptake of macromolecules through the paracellular space	Baricitinib, Tofacitinib	[8,9,66]
STAT3 and JAK2	Leptin	Downregulation of angulin-1	Tricellulin localization is altered; Increased intestinal permeability	Stattic, WP1006, and partially by AG490	[66,67,68]
Undetermined	IFN-γ	Downregulation and redistribution of ZO-1	Increased paracellular permeability to macromolecules	Tofacitinib	[8,9]

## Data Availability

Not applicable.

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
