# Peer review of "JAK-STAT Pathway Regulation of Intestinal Permeability: Pathogenic Roles and Therapeutic Opportunities in Inflammatory Bowel Disease"

_pharmaceuticals, 2021, doi:10.3390/ph14090840_

Round 1
Reviewer 1 Report
This is generally a well written and comprehensive review, but too long and complicated, and therefore disrupting the attention from the main focus: “JAK-STAT Pathway in IBD”. Although very detailed, this is not a review about intestinal permeability and factors involved in maintaining the intestinal barrier. Main comments:
- Title of the review includes “Pathogenic Roles”. However, nothing about this topic is mentioned in the Abstract. Just about the JAK inhibitors for therapy, which is well stated. Please add.
- Abstract: please include the missing info about the pathogenic roles.
- Introduction:
- Instead of references 1-4 (which are way too old), please use the recent guidelines, as you have a lot of choices: AGA Clinical Practice Guidelines – Crohn’s disease - 2021, ulcerative colitis – 2020; ACG Clinical Guideline – Crohn’s disease – 2018, ulcerative colitis – 2019; ECCO guidelines – Crohn’s disease – 2020, ulcerative colitis – 2017; ECCO-ESGAR guideline in IBD – 2018. Please also correct and update info in lines 20-29, according to the mentioned references (e.g., intestinal ulcers are not symptoms etc etc).
- Lines 32-33: The authors wrote: “Despite the abundance of therapeutic strategies for IBD, most treatments alleviate symptoms…”. Please revise, as this is not correct. Many types of medication induce mucosal healing (in both UC and CD) and even transmural healing in CD (in a proportion of patients) and not only alleviate symptoms.
- 9 is old. Please update.
- Lines 140-142 – “Nucleotide-binding oligomerization domain containing protein 2 (NOD2) was the first susceptibility gene that was identified to be associated with CD and plays a major role as an intracellular innate immune sensor”. Here, the first original authors should be listed, not a later reference (Please add Hugot and Ogawa).
- The following references should also be included: “Cleynen I, et al. Genetic factors conferring an increased susceptibility to develop Crohn’s disease also influence disease phenotype: results from the IBDchip European Project. Gut. 2013;62(11):1556-1565.”, “Jostins L, Ripke S, Weersma RK, et al. Host-microbe interactions have shaped the genetic architecture of inflammatory bowel disease. Nature. 2012;491(7422):119-124”. They are crucial, but neglected by these authors. Instead, they cited reviews of the original papers.
- I strongly advise the authors to revise all references in the whole manuscript, in order to make sure that they cited the original papers, not later reviews, as this is not correct.
- Reference 11 is also too old (2007). Please update.
- Also, many other references are too old (except for those regarding JAK-inhibitors), while more recent data are available. Please update, including gut microbiota (positive and negative associations). Fortunately, the authors inserted references regarding findings from the GEM Project (related to intestinal permeability) – here and later on, in their review.
- Aim of the review: Please rephrase it, so that it appears clearer regarding the title of the review. Or, adapt the title.
- Intestinal Permeability in IBD
- Nicely structured; please make sure you insert the most recent references.
- Figure 1 – please explain the name of all the cells, in the part “Under inflammation”. For specialists, it is easy, but not for those who are eager to learn.
- Pore Pathway: Electrolyte Flux:
- I do not see the point of having a sub-paragraph 4.1. Claudins. There is no sub-paragraph 4.2.
- Also, I would advise the authors to focus on their topic: ”JAK-STAT Pathway Regulation of Intestinal Permeability: Pathogenic Roles and Therapeutic Opportunities in IBD” and remove any extra data regarding claudins (other organs and systems etc).
- Figure 2: Please explain all abbreviations. Please add all explanations of the signs in the part “Intestinal inflammation with JAK inhibitors”. I also would advise to name this part of Figure 2: “Intestinal inflammation and roles of JAK inhibitors”
- Leak Pathway: Molecular Mediators
- 1. Zonula Occludens – Similarly, I would suggest to focus on the relationship between JAK-STAT signaling pathway and IBD (which is well written), and to remove extra-info. The data presented by the authors are interesting, but they appear too exhaustive.
- Figure 3: Please explain all abbreviations. Please add all explanations of the signs in the part “Intestinal inflammation with JAK inhibitors”. I also would advise to name this part of Figure 3: “Intestinal inflammation and roles of JAK inhibitors”
- Occludin – same as above. Please remove any extra info not related to JAK-STAT and IBD.
- Tricellulin – same as above.
- JAM-A – same as above.
- Table 1. Please define all abbreviations.
- The authors chose to write both roles of JAK-STAT signaling pathway in IBD pathogenesis and of JAK—inhibitors in the same paragraphs. These should be separated and emphasized.
- I would suggest to include, in a Table, all ongoing studies regarding JAK-inhibitors in IBD (from clinicaltrials.gov), in order to bring something new.
- Format of all references should be revised and corrected.
Author Response
"Please see the attachment."

Reviewer 2 Report
Lei et al provide a comprehensive summary of the role of intestinal barrier function/permeability in the pathogenesis of inflammatory bowel disease (IBD), highlight factors such as tight junction proteins and intestinal cell apoptosis in regulating the intestinal barrier, and the effects of JAK-STAT signaling and inhibition on these various pathways. This is overall an excellent and well-written review that would be a great resource and interest to the readers of Pharmaceuticals. I have the following minor critiques and recommendations:
1. Introduction: The introduction is rather long and dense. While it is important for the authors to acknowledge the other factors (genetics, gut microbiome, immune dysregulation) in the pathogenesis in IBD, the authors should be more concise since this review is focused on intestinal barrier function. I suggest removing the extraneous details about the other IBD-related factors and proceed straight to introducing the topic of intestinal barrier function. Furthermore, the authors should acknowledge that while therapies in IBD are focused on attenuating inflammatory pathways, there are no specific IBD therapies approved or in clinical practice focused on protecting/regenerating the intestinal barrier.
2. Pore Pathway: Electrolyte Flux (Page 6-7): IL-22 is an interesting cytokine and may have "double-edge" effects in the pathogenesis of IBD. IL-22 can promote inflammatory pathways through downstream signaling of JAK-STAT but also promotes intestinal-stem-cell mediated epithelial regeneration (Lindemans, C.A., Calafiore, M., Mertelsmann, A.M., O’connor, M.H., Dudakov, J.A., Jenq, R.R., Velardi, E., Young, L.F., Smith, O.M., Lawrence, G. and Ivanov, J.A., 2015. Interleukin-22 promotes intestinal-stem-cell-mediated epithelial regeneration. Nature, 528(7583), pp.560-564). Is anything known about how the JAK-STAT signaling pathway (and JAK-STAT inhibition) interacts with IL-22 (directly and downstream in terms of its proinflammatory and epithelial protective functions)?
3. JAK-STAT Signaling in Apoptosis and Necroptosis of Intestinal Epithelial Cells (Page 12): Stem cells play critical roles in intestinal epithelial regeneration after inflammatory insult. The authors report "Notably, the JAK-STAT1 pathway was found to be necessary for activation of reserve stem cells during inflammation." Although JAK-STAT signaling inhibition (e.g. Tofacitinib) may attentuate inflammation, does JAK-STAT inhibition attenuate the ability to activate stem cell reserves during inflammation for epithelial regeneration? In this scenario, JAK-STAT inhibition may be detrimental to epithelial barrier recovery.
Author Response
"Please see the attachment."

Round 2
Reviewer 1 Report
I am very pleased with this new version of the manuscript. The authors accepted the importance of the reviewer’s comments and made all adjustments. Now, this review looks tidy, clear and well organized, as well as up-to-date. Figures are provided with all the necessary data. It is a big plus now that the authors inserted the ongoing clinical trials on JAK-inhibitors in IBD patients, as suggested (Table 1). I support the publication of this manuscript.